

# Searching for best lower dimensional visualization angles for high dimensional RNA-Seq data

Wanli Zhang* and Yanming Di*

Department of Statistics, Oregon State University, Corvallis, OR, USA
* These authors contributed equally to this work.

## ABSTRACT

The accumulation of RNA sequencing (RNA-Seq) gene expression data in recent years has resulted in large and complex data sets of high dimensions. Exploratory analysis, including data mining and visualization, reveals hidden patterns and potential outliers in such data, but is often challenged by the high dimensional nature of the data. The scatterplot matrix is a commonly used tool for visualizing multivariate data, and allows us to view multiple bivariate relationships simultaneously. However, the scatterplot matrix becomes less effective for high dimensional data because the number of bivariate displays increases quadratically with data dimensionality. In this study, we introduce a selection criterion for each bivariate scatterplot and design/implement an algorithm that automatically scan and rank all possible scatterplots, with the goal of identifying the plots in which separation between two pre-defined groups is maximized. By applying our method to a multi-experiment *Arabidopsis* RNA-Seq data set, we were able to successfully pinpoint the visualization angles where genes from two biological pathways are the most separated, as well as identify potential outliers.

## INTRODUCTION

High throughput RNA sequencing (RNA-Seq) has been widely adopted for quantifying relative gene expression in comparative transcriptome analysis. In recent years, the increasing number of RNA-seq studies on the model plant *Arabidopsis thaliana* have resulted in an ever-accumulating amount of data from multiple RNA-Seq experiments. In this article, we will develop tools for the exploration and visualization of such multi-experiment data.

For examining treatment effects of individual genes under multiple conditions and across multiple experiments, a vector summarizing the differential expression (DE) results under different treatment conditions seems adequate. To visualize the DE profile under different treatments, a line plot can be used. However, since genes work interactively in all biological processes, it is of interest to examine expression patterns of groups of genes, through which the genes' biological context can be better understood. In light of this, researchers often would like to both identify the general trend and pinpoint individual

Corresponding author
Wanli Zhang,
zhangwa@stat.oregonstate.edu

aberrations in the expression profile of genes belonging to the same biological pathway, as well as compare the profiles between multiple pathways.

When multiple genes are being examined together, the line plots are less effective for visualizing DE or expression profiles: The lines often cross each other, making it difficult to identify the grouping and understand the behavior of individual genes. One common alternative visualization method is the scatterplot, which shows expression level under two treatment conditions at a time. Scatterplots are effective in showing clustering patterns and outliers, greatly assisting with data exploration (*Elmqvist, Dragicevic & Fekete, 2008*). For high dimensional data, one has the option of using the scatterplot matrix, in which each panel is the scatterplot for the corresponding pair of features. However, manual scanning of all possible pairwise scatterplots can be arduous or even fruitless at times, because the number of possible visualization angles increases quadratically with respect to data dimensionality ($\binom{p}{2}$ possible angles).

In this paper, we propose to automatically search for the best low dimensional visualization angles (two-, three-, or four-dimensional) based on a context-sensitive, numeric measure of importance, thereby reducing the amount of effort invested in scatterplot scanning. In our study, we hope to explore the patterns and differences in gene expression profile between two phytohormone signaling pathways, and therefore, we would like the top ranked scatterplots to contain as much information as possible on pathway classification. We thus define such an importance measure for the dimension subsets such that the scatterplots will show the largest separation between different pre-defined groups in the data set.

For this study, we will look for feature subsets upon which the pathways ethylene (ET) and jasmonic acid (JA) are the most separated, and quantify the between-group separation by calculating the repeated cross-validation (RCV) error of misclassification using MclustDA (*Fraley & Raftery, 2002*), a model-based classification method. In Fig. 1, we show one of the top ranked 2-subset feature combinations that give the greatest separation between two pathways, as well as subset giving the smallest separation. Comparing the two scatterplots, we can observe that the two pathway groups in Fig. 1A are more visually distinguishable than those in Fig. 1B.

The rest of the paper is formatted as follows: "Data Description and Processing" outlines the collection and processing of the data and information on the experiments and biological pathways. The statistical methods are described in "Methods." In "Results," we list the results obtained by applying our method to the collected data. Finally, we state our conclusion and discuss limitations and possibilities for future work in "Conclusion." Additional proofs and graphs are included in the Appendix.

## DATA DESCRIPTION AND PROCESSING

### Collecting experimental data

In this study, we use a portion of the data collected and processed by *Zhuo et al. (2016)* The original data were acquired from the National Center for Biotechnology Information website (www.ncbi.nlm.nih.gov) and processed through a customized assembly pipeline to obtain a matrix of counts for genes in samples. All datasets originate from RNA-Seq
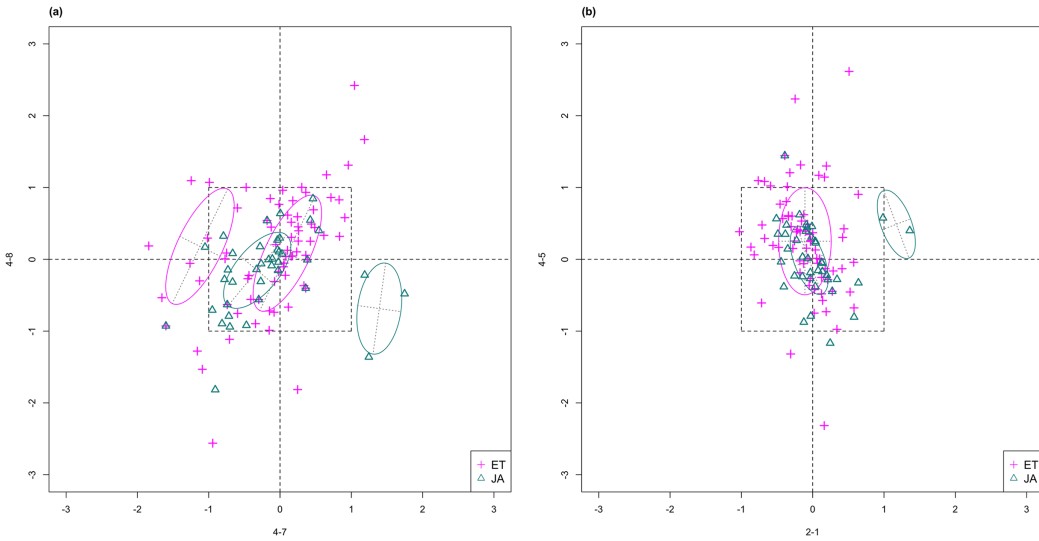

**Figure 1 Scatterplots of two-dimensional feature subsets reflecting maximum (A) and minimum (B) group separations.** Dashed-line square marks ±1 range from the origin. Different classes distinguished with color. Ellipses correspond to component mean and covariance fitted by MclustDA. Treatment *i-j* represents the *j*th treatment-control contrast in experiment *i*.

experiments on the model plant *A. thaliana*, with treatment conditions (including genetic variants) varying between experiments. Tissue types in the original data include leaf, seed, and multi-tissue. The number of treatments/factor levels also vary among the experiments.

For this article, we will focus on experiments conducted on the leaf tissue of *Arabidopsis*, which include a total of five datasets. The Gene Expression Omnibus (GEO) accession numbers (which can be used to directly search for the experiment/dataset information) are available as part of the meta-data, and the assembly pipeline that produced the read count matrix is described in detail in (*Zhuo et al. 2016*). We have included some basic information on the experiments in Table 1.

## Estimating log fold changes

Let $c_{ij}$ be the raw count of RNA-Seq reads mapped to gene $i$ in biological sample $j$. For each gene $i$, to assess gene expression, we often consider the relative read frequencies $c_{ij}/N_j$ where $N_j$ is the total number of reads mapped to sample $j$. $N_j$ can vary greatly across samples. A more subtle issue is that $N_j$ can be significantly influenced by a few genes with extremely high read counts. To address this issue, *Anders & Huber (2010)* proposed to normalize $N_j$'s by multiplying them with so-called normalization factors $R_j$'s. Using their method, $R_j$'s are estimated such that the median fold change (over all genes $i$) will be one between each column of normalized relative frequencies, $c_{ij}/N_j R_j$, and a pseudo column, $c_{i0}/N_0 R_0$ (for each gene $i$, $c_{i0}$ is the geometric mean of $c_{ij}$ over all samples $j = 1, ..., n$, $N_0$ is the column total of $c_{i0}$, and see *Zhuo et al. (2016)* for a formula for $R_0$ and other details). In *Zhuo et al. (2016)*, we identified a set of 104 genes that were shown to be relatively stably expressed across all biological samples in a collection of *Arabidopsis* RNA-Seq experiments (including the five experiments we use in this study). In this paper, we used these 104 genes as a reference gene
**Table 1 Experiment information.**

| ID | GEO accession # | Title | Platform | Sample size |
|---|---|---|---|---|
| 1 | GSE36626 | Dynamic deposition of the histone H3.3 variant accompanies developmental remodeling of Arabidopsis transcriptome (mRNA-Seq) | GPL11221 *Illumina Genome Analyzer IIx* | 4 |
| 2 | GSE39463 | Time-course RNA-seq analysis of the barley MLA1 immune receptor-mediated response to barley powdery mildew fungus Bgh in *Arabidopsis thaliana* | GPL13222 *Illumina HiSeq 2000* | 48 |
| 3 | GSE48235 | Four distinct types of dehydration stress memory genes in *Arabidopsis thaliana* | GPL9302 *Illumina Genome Analyzer II* | 6 |
| 4 | GSE51304 | Non-CG methylation patterns shape the epigenetic landscape in *Arabidopsis* | GPL13222 *Illumina HiSeq 2000* | 18 |
| 5 | GSE54677 | Transcriptional gene silencing by Arabidopsis microchidia homologues involves the formation of heteromers | GPL13222 *Illumina HiSeq 2000* | 20 |

set to compute the normalization factors. With the normalization factors $R_j$ estimated this way, the median fold change in normalized relative frequencies over these 104 genes will be one between any sample $j$ and the pseudo sample.

With these estimated normalization factors $R_j$, we fit a log-linear negative binomial regression model to each row $i$ (one gene) of the read count matrix (separately for each experiment) to assess DE between treatment groups (we suppressed the subscript $i$ in the model equation below):

$$c_j \sim \text{Negative Binomial}\ (\mu_j, \phi),$$
$$\log(\mu_j) = \log(N_j R_j) + x_{j0}\beta_0 + x_{j1}\beta_1 + \dots + x_{jp}\beta_p,$$

for $j = 1, \dots, n$, where $\phi$ is the overdispersion parameter capturing the commonly observed extra-Poisson variation in the read count data, and $(x_{jk})$ is the model matrix. We chose the model matrix such that the fitted value of $\beta_0$ will correspond to the log mean relative count of the control group, and the fitted regression coefficients $\beta_1, \dots, \beta_p$ will correspond to log fold changes between a treatment group and the control group. We used our own NBPSeq package to fit such a log-linear regression model (*Di, 2015*). These estimated log fold changes (converted to base 2) from the five experiments described in "Collecting experimental data" are the feature sets used in this paper.

## Finding pathway genes

For this study, we focus our attention on the signaling pathways of two phytohormones: ethylene (ET) and jasmonic acid (JA). As a plant hormone, ET is commercially important due to its regulation on fruit ripening (*Lin, Zhong & Grierson, 2009*). JA acts as a key cellular signal involved in the activation of immune responses to most insect herbivores and necrotrophic microorganisms (*Ballaré, 2010*).

For each pathway, we first use AmiGO 2 (http://amigo.geneontology.org/amigo/landing) to search for the list of genes involved, and then identify the subset of genes in our data set that belong to the pathway through cross-reference. Genes with a fold change of <2 under all treatment conditions are filtered out. The name, GO accession number, and the number of genes in each pathway are listed in Table 2.

| Table 2 Pathway information. | | | |
|---|---|---|---|
| ID | Pathway name | GO accession # | # Genes |
| ET | Ethylene-activated signaling pathway | GO:0009873 | 86 |
| JA | Jasmonic acid mediated signaling pathway | GO:0009867 | 48 |

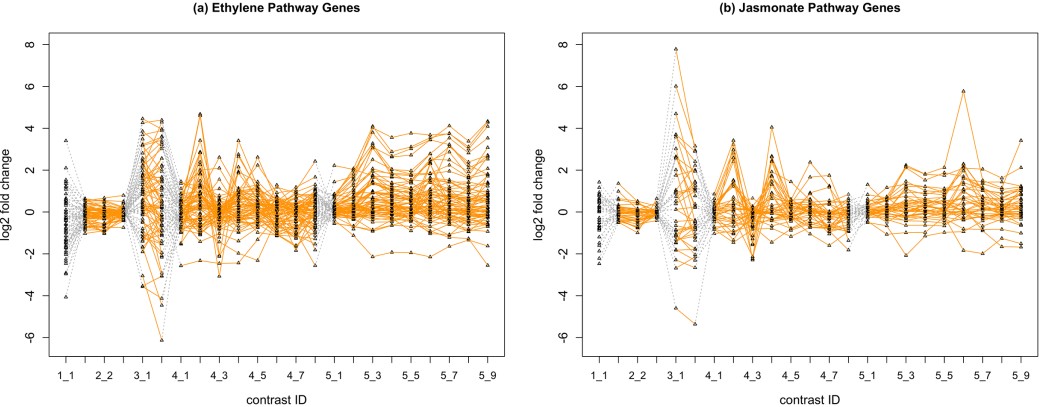

**Figure 2 Gene expression profile plot for pathways ET (A) and JA (B).** Treatments from the same experiment are joined by orange lines. Different experiments are joined by gray dashed lines. Feature *i-j* represents the *j*th treatment-control contrast in experiment *i*.

In Fig. 2, we display the expression profile of genes that belong to each pathway group. Under certain individual treatment-control contrasts (e.g., 2-3, 4-3, 5-1), there exist observable similarities between the distribution of expression levels, while it is more difficult to tell under other treatments.

# METHODS

## Mixture discriminant analysis via MclustDA

In this section, we will start by introducing a classification method named MclustDA, and then define a measure for group separation using cross-validation (CV) results with MclustDA. Finally, we lay out our strategy for reducing data dimensionality with the ultimate goal of simplifying navigation of scatterplots.

### MclustDA model

In discriminant analysis (DA), known classifications of some observations are used to classify others. The number of classes, $G$, is assumed to be known. For probabilistic DA methods, it is assumed that observations in class $k$ follow a class specific probability distribution $f_k(\cdot)$. Let $\tau_k$ represent the proportion of observations in class $k$. According to Bayes's theorem, it follows that

$$P(\boldsymbol{y} \in \text{class } j) = \frac{\tau_j f_j(\boldsymbol{y})}{\sum_{k=1}^{G} \tau_k f_k(\boldsymbol{y})},$$

where observation $\boldsymbol{y}$ is assigned to the most probable class.
Commonly used DA methods, including Fisher's linear discriminant analysis (LDA) and quadratic discriminant analysis (QDA), assume a multivariate normal density for each class:

$$f_k(\boldsymbol{y}) = \phi(\boldsymbol{y}|\mu_k, \Sigma_k).$$

The method is called LDA if the covariance matrices for all classes coincide ($\Sigma_k = \Sigma$ for $k = 1, ..., G$), and is called QDA if the class covariances are allowed to vary.

MclustDA (*Fraley & Raftery, 2002*), an extension and generalization to LDA and QDA, models each class density as a mixture of multivariate normals. The density for class $j$ is as follows:

$$f_j(\boldsymbol{y}|\theta_k) = \sum_{k=1}^{G_j} \tau_{jk}\phi(\boldsymbol{y}|\mu_{jk}, \Sigma_{jk}),$$

where $G_j$ is the number of components for class $j$, $\{\tau_{jk}\}$ are mixing proportions for components in class $j$, and $\theta_k$ is the vector of parameters for the normal mixture. Component covariances $\Sigma_{jk}$ are allowed to vary both within and between classes.

Parameters within each class are separately estimated by maximum likelihood (ML) via the EM algorithm (*Dempster, Laird & Rubin, 1977*), which is equivalent to fitting a Mclust (*Fraley & Raftery, 2002*) model for each class. Just like Mclust, MclustDA performs model selection within each class for the number of mixture components as well as covariance matrix parameterizations with Bayesian information criterion (*Schwarz, 1978*).

### Comparison with LDA

In our study, MclustDA is chosen over LDA/QDA as the classifier due to its greater flexibility in describing the data. In RNA-Seq analysis, we typically assume that the majority of genes are not differentially expressed, and therefore we expect to see a cluster of points around the origin. Since MclustDA proposes to fit more than one normal component to each class, it readily captures the cluster of non-DE genes as well as any abnormalities that might be of interest.

In Fig. 3, we fitted a MclustDA model and a LDA model on dimensions [3-1, 4-2] of our data, separately. In MclustDA fit, each class is described with a mixture of two bivariate normal components, with the ellipses representing fitted covariance matrix estimates. For details in how the ellipses are constructed, see Appendix A.

Class JA is fitted with a component centered near the origin, representing genes with low expression levels under both treatments, as well as a component centered at (2.276, 1.663) that encompasses relatively active genes. Class ET is represented by a single normal component centered at (0.537, 0.406).

In comparison, due to model assumptions, LDA fitted a bivariate normal density to each class with covariances being equal, and in this case, the estimated centers almost coincide with each other. The fitted normal densities are only able to capture the general shape and orientation of each class, while MclustDA provides us with a more detailed anatomy of geometric and distributional properties in each class.

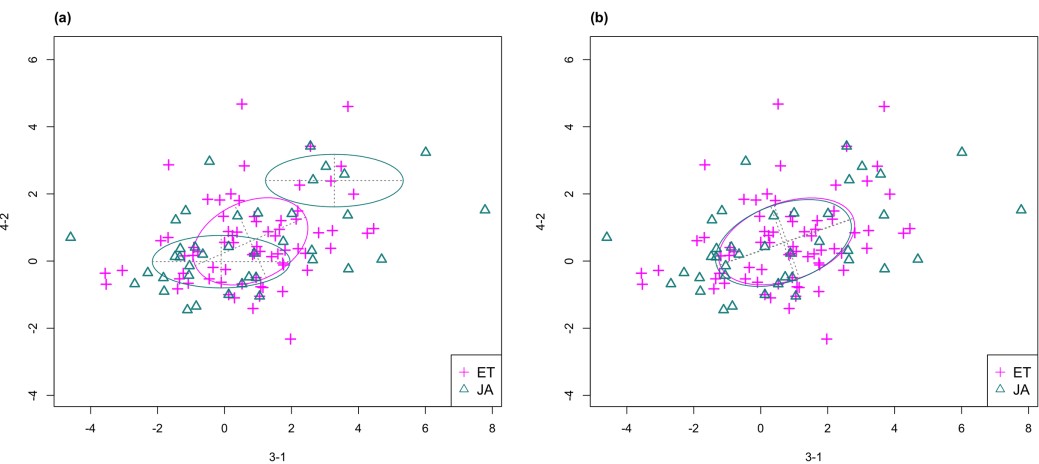

**Figure 3 Comparison of MclustDA (A) and LDA (B) fit of the same data.** Fitted components and points from different classes are distinguished with color. Ellipses correspond to component covariances.

## Quantification of group separation

Our definition of group separation measure is motivated by the relationship between visualized separation and misclassification probability (from a model-based classifier).

Suppose we wish to separate two populations $\pi_1$ and $\pi_2$. Let $X = [X_1, ..., X_p]$ denote the $p$-dimensional measurement vector of an observation. We assume that densities $f_1(x)$ and $f_2(x)$ describe the variability of the two populations. Let $p_1$ and $p_2$ denote prior probability of each population. Define $c(1|2)$ and $c(2|1)$ as costs of misclassifying an object from class $2(1)$ as class $1(2)$. Here we let $c(1|2) = c(2|1) = 1$ to simplify the formulation. Let $\Omega$ denote the entire sample space, and $\Omega = R_1 \cup R_2$, where $R_1$ is the set of values of $x$ for which we classify objects into $\pi_1$, and $R_2 = \Omega - R_1$.

The probability of misclassifying an object from $\pi_1$ as $\pi_2$ is:

$$P(2|1) = P(X \in R_2|\pi_1) = \int_{R_2} f_1(x)\mathrm{d}x,$$

and similarly, we have

$$P(1|2) = P(X \in R_1|\pi_2) = \int_{R_1} f_2(x)\mathrm{d}x.$$

By definition, we can calculate the probability of misclassifying any object:

$$P(\text{misclassified as } \pi_1) = P(X \in R_1|\pi_2)P(\pi_2) = P(1|2)p_2,$$
$$P(\text{misclassified as } \pi_2) = P(X \in R_2|\pi_1)P(\pi_1) = P(2|1)p_1.$$

The total probability of misclassification (TPM) is defined as the probability of either misclassifying a $\pi_1$ object or misclassifying a $\pi_2$ object, that is,

$$\text{TPM} = p_1 P(2|1) + p_2 P(1|2). \tag{1}$$
**Table 3 Confusion matrix.**

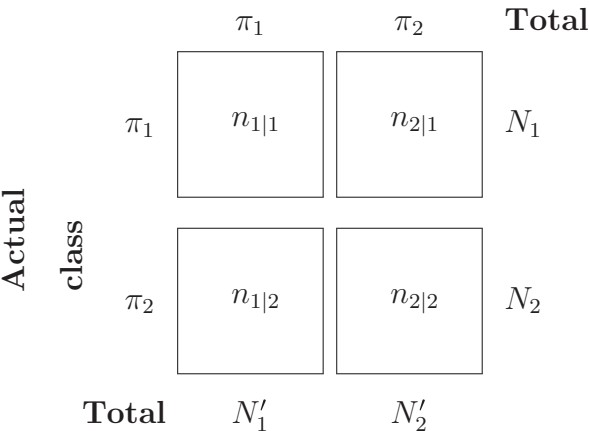

Suppose $Y = \{Y_1, ..., Y_{N1}\} \sim \pi_1$ and $Z = \{Z_1, ..., Z_{N2}\} \sim \pi_2$ are two i.i.d samples from the two populations. Assume that a classification system has been trained and tested on this data set, and results in the confusion matrix in Table 3.

Then the misclassification error rate (MER), that is, probability of misclassifying any object, is given by:

$$\text{MER} = \frac{n_{1|2} + n_{2|1}}{N_1 + N_2} = \frac{n_{1|2}}{N_2} \cdot \frac{N_2}{N_1 + N_2} + \frac{n_{2|1}}{N_1} \cdot \frac{N_1}{N_1 + N_2}. \tag{2}$$

Under the assumption that each object is independently classified, the number of misclassified $\pi_1$ objects, $N_{2|1}$, follows a Binomial distribution with parameters $(N_1, P(2|1))$. Likewise, the number of misclassified $\pi_2$ objects, $N_{1|2}$, follows a Binomial distribution with parameters $(N_2, P(1|2))$. The ML estimators for $P(2|1)$ and $P(1|2)$ can be easily computed:

$$\widehat{P(2|1)} = \frac{n_{2|1}}{N_1}; \quad \widehat{P(1|2)} = \frac{n_{1|2}}{N_2}.$$

Now, if we set $p_1 = N_1/(N_1 + N_2)$ and $p_2 = N_2/(N_1 + N_2)$ as prior probabilities for $\pi_1$ and $\pi_2$, then under independence assumption, it follows that

$$\text{MER} = p_1 \widehat{P(2|1)} + p_2 \widehat{P(1|2)},$$

that is, MER is a maximum likelihood, and hence consistent, estimate of TPM.

In practice, however, the MER tends to underestimate TPM because the same data has been used for both training and testing. In this study, we use CV to address this issue.

### Repeated stratified cross-validation

One of the most commonly used method to estimate the expected error rate is CV. For a $K$-fold CV, the original data is randomly split into $K$ equally sized subsamples, of which $K-1$ (training set) are used to train a classifier and the remaining one (validation set) is used to test the trained classifier. For a binary classification problem, the MER, as defined in (2), is typically computed using the validation set as a performance measure for the classifier. The training-validation process is iterated over all $K$ folds, each time using a different subsample as validation set, and the resulting $K$ MER values are averaged. In stratified CV, the folds are selected so that they contain approximately the same proportion of classes as the original data. It has been shown in previous studies that stratified CV tends to perform uniformly better than CV, in terms of both bias and variance (*Kohavi, 1995*).

Due to the randomness in partitioning the sample into $K$ folds, we have introduced variation into the $K$-fold CV estimator. One way to reduce this variation is to repeat the whole CV process multiple times using different pseudorandom allocations of instances to training and validation folds for each repetition (*Kim, 2009*), and report the average of CV estimators across all repetitions. This method is often referred to as the RCV. For improved repeatability of results, common seeding has been recommended in earlier studies (*Powers & Atyabi, 2012*). In our implementation, we set a fixed random number seed for each repetition of CV.

Let $C \times K$-CV denote a $K$-fold CV with $C$ repetitions. There has been much discussion on the optimal choice of $C$ and $K$ (*Kohavi, 1995*; *Kim, 2009*; *Powers & Atyabi, 2012*). Increasing $C$ tends to decrease the variance of the RCV estimator, but at the same time increases the computational time. The choice of $K$ takes into account the tradeoff between bias and variance of the CV estimator (of the expected error rate). For small $K$, less data is used to train the classifier and therefore the error estimate tends to be biased. For large $K$, the estimator becomes less biased due to more data being used in training, but its variance is inflated due to higher correlation between different training folds. *Kohavi (1995)* recommends using a stratified 10-fold CV with multiple runs, and we chose $C = 10$ considering the amount of computation required as well as the specs of our hardware.

### Quantify group separation

We define the group separation index (GSI) as

$$\text{GSI} = 1 - \hat{\epsilon}_{\text{rcv}}, \tag{3}$$

where $\hat{\epsilon}_{\text{rcv}}$ denotes the repeated stratified CV estimator of the total misclassification probability using MclustDA as the classifier.

Intuitively, for a chosen feature subset, a small CV error indicates that the data can be more easily classified when projected onto these dimensions, which, in our expectation, can be reflected in the graphical representation of the data by showing that different classes can be more easily distinguished through simple visualization.

## Feature subset selection via GSI ranking

In this section, we describe the data in each pathway with a low dimensional representation for easier interpretation by selecting a parsimonious subset of features (treatment-control contrasts) that contain as much information on pathway classification/separation as possible. In other words, we hope to find the dimensions to project the data onto such that the separation between two pathways is as large as possible. We use GSI, as defined in (3), to measure the separation between two pathway groups.

In order to find the optimal feature subset in terms of group separation, we designed and implemented the following algorithm:

**Step 1**: Determine the number of features $M$ to keep. Choose $M$ from {2, 3, 4}.

**Step 2**: List all $M$-subsets of features exhaustively. Call this collection of subsets $\mathcal{F}_M$.

**Step 3**: For each member of $\mathcal{F}_M$, subset the data accordingly. Calculate and record a $10 \times 10$ stratified CV error rate (and equivalently, GSI) with MclustDA as classifier on each subsetted data. For each fold of CV, use MER as measure of fit.

- CV model fitting: First fit a MclustDA model to the entire subsetted data, setting maximum number of components as $G_j \equiv G = 2$. Then, use the same fitted model (number of components, covariance parameterization) for every fold of CV.

**Step 4**: Rank the feature subsets in $\mathcal{F}_M$ according to their GSI values. Feature subsets with higher GSI values are ranked higher.

**Step 5**: Repeat above steps for other values of $M$.

Concerning the choice of maximum number of components in Step 3, we initially attempted to use the default value $G = 5$ in R function `MclustDA`, and discovered that in some situations the number of model parameters became too large for the algorithm to produce a meaningful point estimate. The same problem occurred for $G = 4$ and $G = 3$, especially when we looked at three- and four-dimensional data. Therefore, we settled on using $G = 2$ for our implementation.

For the purpose of finding "good" angles for data visualization, we will examine the scatterplots and scatterplot matrices generated by top-ranked feature subsets. The results will be discussed in "Results."

### Random number seed

To ensure reproducibility of our results, for each of 2-, 3- and 4-subset selection process, we followed the following protocol to set random number seeds:

**Step 1**: Choose a list of 50 random number seeds. Partition the list into five batches of 10 seeds.

**Step 2**: For each feature subset, run 10-fold stratified CV for 50 times, each time using a different seed from the list.
**Step 3**: Average results within each of five batches of 10 random seeds to obtain $10 \times 10$ stratified CV result. For instance, average of seeds 1–10 results serves as first run of $10 \times 10$ RCV; average of seeds 11–20 serves as second run, etc.

### GSI ranking with LDA as classifier

As discussed in "Mixture discriminant analysis via MclustDA," LDA fits only one multivariate normal component to observations in each class. In gene expression data such as ours, a majority of genes are expected to be expressed at a low level, meaning we are likely to observe a cluster of data points around the origin, regardless of the class they belong to. Meanwhile, any aberrant patterns demonstrated by individual data points are often not captured by fitting LDA model.

To see whether our method can still discover interesting visualization angles with LDA as classifier, we modified and implemented the GSI calculation algorithm accordingly, and applied it to our data. Results are presented in "Using fisher LDA as classifier."

### Dimension reduction via PCA

Principal component analysis (PCA) maps the data onto a lower dimensional space in such a way that the variance of the data in the low-dimensional representation is maximized. As a dimension reduction technique, usually only the first few principal components (PCs) are used. Despite its popularity in the field of data visualization, the formulation of PCA does not involve any class information in the data, which implies that the projected directions corresponding to the largest variance may not contain the best separability information.

To verify this observation, using the expression data from all five experiments, we calculated its PCs, and treat them as the new (projected) features. Then for the first two, three and four PCs, respectively, we calculated the GSI for each case using $10 \times 10$-CV with MclustDA and compare the results with ours.

## RESULTS

### Repeated cross-validation with MclustDA

With the secondary purpose of testing the stability of repeated CV, we executed multiple runs for each of the 2-, 3-, and 4-subset feature selection procedures. The top ranked feature subsets as well as their corresponding GSI values are presented in Tables 4–6.

#### Stability of RCV model selection results

Although the top ranked feature subsets sometimes differ between multiple RCV runs, we are still able to observe high degree of overlap between the results:

For 4-subset (Table 6), [1-1, 4-2, 4-4, 5-2], [3-1, 4-4, 4-5, 5-2] and [3-2, 4-5, 4-7, 5-7] are among top ranked feature combinations in all five runs.

For 3-subset (Table 5), feature combinations [3-2, 4-7, 5-2], [3-2, 5-2, 5-6] and [1-1, 3-1, 5-2] are ranked top for all five runs.

For 2-subset (Table 4), [4-7, 4-8], [3-1, 3-2], [4-7, 5-2] and [3-1, 5-2] are among top ranked feature combinations for all runs.
**Table 4 Top ranked 2-subsets from multiple runs of 10 × 10 RCV.**

**(a) Run 1**

| Rank | Subset | GSI |
|---|---|---|
| 1 | [4-7, 4-8] | 0.698 |
| 2 | [3-1, 3-2] | 0.694 |
| 3 | [4-7, 5-2] | 0.688 |
| 4 | [3-1, 5-2] | 0.681 |
| 5 | [2-1, 4-7] | 0.680 |

**(b) Run 2**

| Rank | Subset | GSI |
|---|---|---|
| 1 | [4-7, 4-8] | 0.703 |
| 2 | [3-1, 3-2] | 0.696 |
| 3 | [4-7, 5-2] | 0.690 |
| 4 | [3-1, 5-8] | 0.682 |
| 5 | [3-1, 5-2] | 0.681 |

**(c) Run 3**

| Rank | Subset | GSI |
|---|---|---|
| 1* | [3-1, 3-2] | 0.697 |
| 2* | [4-7, 5-2] | 0.697 |
| 3 | [4-7, 4-8] | 0.689 |
| 4 | [3-1, 5-2] | 0.685 |
| 5 | [2-1, 4-7] | 0.675 |

**(d) Run 4**

| Rank | Subset | GSI |
|---|---|---|
| 1 | [3-1, 3-2] | 0.704 |
| 2 | [4-7, 4-8] | 0.695 |
| 3 | [3-1, 5-2] | 0.687 |
| 4 | [3-1, 5-8] | 0.684 |
| 5 | [4-7, 5-2] | 0.681 |

**(e) Run 5**

| Rank | Subset | GSI |
|---|---|---|
| 1 | [4-7, 4-8] | 0.708 |
| 2 | [3-1, 3-2] | 0.702 |
| 3 | [4-7, 5-2] | 0.689 |
| 4 | [4-4, 5-2] | 0.688 |
| 5 | [3-1, 5-2] | 0.683 |

**Note:**
Ties are marked with asterisk (*). Combinations appearing in all five runs are highlighted with distinguishing colors.

### Top ranked scatterplot: same experiment

In Fig. 4, we show the scatterplot of the data projected onto dimensions [4-7, 4-8], one of the top ranked 2-subset feature combinations. These two features originate from the same experiment.

**Table 5 Top ranked 3-subsets from multiple runs of 10 × 10 RCV.**

**(a) Run 1**

| Rank | Subset | GSI |
|---|---|---|
| 1 | [3-2, 4-7, 5-2] | 0.724 |
| 2 | [3-2, 5-2, 5-6] | 0.720 |
| 3 | [1-1, 3-1, 5-2] | 0.718 |
| 4 | [3-1, 5-2, 5-6] | 0.710 |
| 5 | [2-2, 2-3, 3-1] | 0.707 |

**(b) Run 2**

| Rank | Subset | GSI |
|---|---|---|
| 1* | [1-1, 3-1, 5-2] | 0.717 |
| 2* | [3-2, 4-7, 5-2] | 0.717 |
| 3 | [3-2, 5-2, 5-6] | 0.713 |
| 4 | [1-1, 2-2, 3-1] | 0.708 |
| 5 | [2-2, 2-3, 3-1] | 0.708 |

**(c) Run 3**

| Rank | Subset | GSI |
|---|---|---|
| 1 | [3-2, 5-2, 5-6] | 0.725 |
| 2 | [1-1, 3-1, 5-2] | 0.717 |
| 3* | [3-1, 5-2, 5-9] | 0.715 |
| 4* | [3-2, 4-7, 5-2] | 0.715 |
| 5 | [3-1, 4-7, 5-2] | 0.714 |

**(d) Run 4**

| Rank | Subset | GSI |
|---|---|---|
| 1 | [3-2, 5-2, 5-6] | 0.736 |
| 2 | [1-1, 3-1, 5-2] | 0.728 |
| 3 | [2-2, 2-3, 3-1] | 0.713 |
| 4* | [3-1, 4-7, 5-2] | 0.712 |
| 5* | [3-2, 4-7, 5-2] | 0.712 |

**(e) Run 5**

| Rank | Subset | GSI |
|---|---|---|
| 1 | [1-1, 3-1, 5-2] | 0.726 |
| 2 | [3-2, 4-7, 5-2] | 0.724 |
| 3 | [3-2, 5-2, 5-6] | 0.720 |
| 4 | [2-3, 3-1, 5-2] | 0.715 |
| 5 | [2-2, 2-3, 3-1] | 0.710 |

**Note:**
Ties are marked with asterisk (*). Combinations appearing in all five runs are highlighted with distinguishing colors.

*Experiment 4*

Since both features originate from the same experiment, we will focus on the context of this experiment and first present some background information. The purpose of Experiment 4 is to characterize non-CG methylation and its interaction with histone

**Table 6 Top ranked 4-subsets from multiple runs of 10 × 10 RCV.**

**(a) Run 1**

| Rank | Subset | GSI |
|---|---|---|
| 1 | [1-1, 4-2, 4-4, 5-2] | 0.730 |
| 2 | [4-4, 4-5, 4-7, 5-8] | 0.728 |
| 3 | [3-1, 4-4, 4-5, 5-2] | 0.727 |
| 4 | [4-1, 4-7, 4-8, 5-1] | 0.725 |
| 5 | [3-2, 4-5, 4-7, 5-7] | 0.724 |

**(b) Run 2**

| Rank | Subset | GSI |
|---|---|---|
| 1 | [1-1, 4-2, 4-4, 5-2] | 0.745 |
| 2 | [3-1, 4-4, 4-5, 5-2] | 0.741 |
| 3 | [2-3, 3-1, 4-4, 4-5] | 0.734 |
| 4 | [2-3, 3-1, 4-7, 5-2] | 0.725 |
| 5 | [3-2, 4-5, 4-7, 5-7] | 0.724 |

**(c) Run 3**

| Rank | Subset | GSI |
|---|---|---|
| 1 | [1-1, 4-2, 4-4, 5-2] | 0.735 |
| 2 | [4-4, 4-5, 4-7, 5-8] | 0.731 |
| 3* | [3-2, 4-5, 4-7, 5-7] | 0.726 |
| 4* | [4-1, 4-7, 4-8, 5-1] | 0.726 |
| 5 | [3-1, 4-4, 4-5, 5-2] | 0.725 |

**(d) Run 4**

| Rank | Subset | GSI |
|---|---|---|
| 1 | [1-1, 4-2, 4-4, 5-2] | 0.739 |
| 2* | [3-1, 4-4, 4-5, 5-2] | 0.731 |
| 3* | [3-2, 4-5, 4-7, 5-7] | 0.731 |
| 4 | [4-4, 4-5, 4-7, 5-8] | 0.723 |
| 5 | [3-1, 3-2, 4-7, 5-2] | 0.722 |

**(e) Run 5**

| Rank | Subset | GSI |
|---|---|---|
| 1 | [1-1, 4-2, 4-4, 5-2] | 0.740 |
| 2 | [3-1, 4-4, 4-5, 5-2] | 0.735 |
| 3* | [3-2, 4-5, 4-7, 5-7] | 0.730 |
| 4* | [2-2, 2-3, 3-1, 4-7] | 0.730 |
| 5 | [3-2, 4-6, 4-7, 5-6] | 0.723 |

**Note:**
Ties are marked with asterisk (*). Combinations appearing in all five runs are highlighted with distinguishing colors.

methylation in *A. thaliana* (*Stroud et al., 2014*). Non-CG methylation is a category of DNA methylation, where methyl groups are added to the DNA molecule, altering its chemical structure and thereby changing its activity. DNA methylation is usually catalyzed by

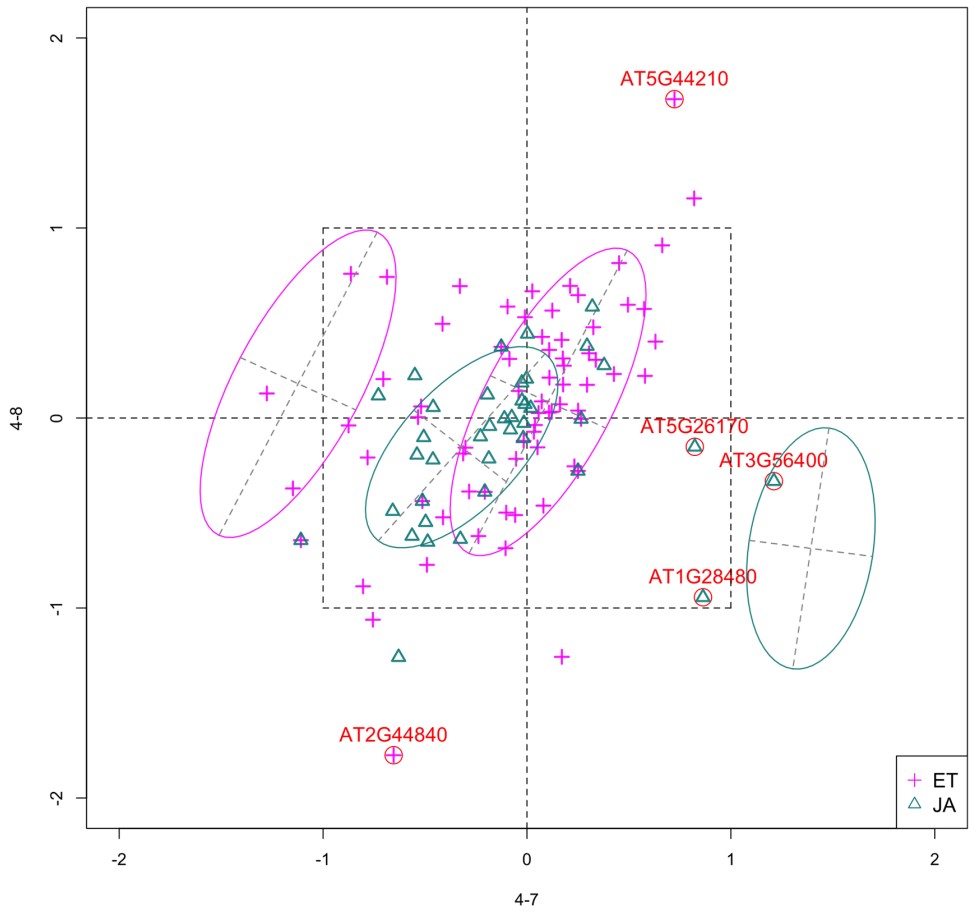

**Figure 4 Scatterplot of data projected on dimensions 4-7 and 4-8.** Pathways are distinguished with color. Ellipses represent estimated covariances fitted by MclustDA. Potential outliers highlighted and labeled with their names. Dashed-line square is ±log(2) range from the origin.

DNA methyltransferases (MTases), which transfer and covalently bind methyl groups to DNA. In *Arabidopsis*, the principal DNA MTases include chromomethylase (CMT) and domains rearranged MTase (DRM) proteins, in particular CMT3 and DRM2. Expression of DRM1 is scarcely detected, while the function of CMT2 has not been studied as well as that of CMT3.

Histone methylation is a process by which methyl groups are transferred to amino acids of histone proteins. Histone methylation can either increase or decrease gene transcription, depending on which amino acids are methylated and the degree of methylation. The methylation process is most commonly observed on lysine residues (K) of histone tails H3 and H4, among which H3K9 (lysine residue at ninth position on H3) serves as a common site for gene inactivation. Lysine methylation requires a specific MTase, usually containing an evolutionarily conserved SET domain. In *Arabidopsis*, Su (var)3–9 homologue 4 (SUVH 4), SUVH 5 and SUVH 6 are the major H3K9 MTases.

Feature 4-7 corresponds to the *drm1 drm2 cmt2 cmt3* quadruple gene knockout mutant, created by crossing *cmt2* to *cmt3* and *drm1 drm2* double mutants. It was found

**Table 7 Feature information.**

| Feature ID | Sample GEO accession # | Description |
|---|---|---|
| 4-0 (control) | GSM1242374, GSM1242375 | Wildtype |
| 4-7 | GSM1242388, GSM1242389 | *drm1 drm2 cmt2 cmt3* quadruple mutant |
| 4-8 | GSM1242390, GSM1242391 | *suvh4 suvh5 suvh6* triple mutant |

**Table 8 Outlier information.**

| Gene name | Description |
|---|---|
| AT5G44210 | Encodes a member of the ERF (ethylene response factor) subfamily B-1 of ERF/AP2 transcription factor family (ATERF-9) |
| AT2G44840 | Same function as AT5G44210; Cell-to-cell mobile mRNA |
| AT5G26170 | WRKY Transcription Factor, Group II-c; Involved in jasmonic acid inducible defense responses. |
| AT3G56400 | WRKY Transcription Factor, Group III; Repressor of JA-regulated genes; Activator of SA-dependent defense genes. |
| AT1G28400 | GATA zinc finger protein |

that non-CG methylation was eliminated in such mutants, indicating that DRM1, DRM2, CMT2 and CMT3 proteins are collectively responsible for all non-CG methylation in *Arabidopsis*. Feature 4-8 corresponds to the *suvh4 suvh5 suvh6* triple mutant. The control group of this experiment corresponds to wildtype *Arabidopsis*. Table 7 summarizes the above information.

### Outliers

Potential outliers from JA pathway, as highlighted and labeled in the scatterplot, fall into the fourth quadrant, which implicates that these genes are up-regulated under 4–7 (DNA methylation) but down-regulated under 4–8 (histone methylation). Information on these genes is collected from TAIR (*Berardini et al., 2015*) and displayed in Table 8. One interesting discovery we made was that one of the outliers, **AT3G56400**, functions as a repressor of JA-regulated genes. In other words, its gene product inhibits the expression of other genes related to JA regulation.

### Pattern differences

The first thing we can observe from the scatterplot is that a majority of genes are expressed at a low level (with fold change <2) under both treatment conditions, as demonstrated by the clustered points inside ±1 square. Although most genes are expressed at a relatively low level, we are still able to identify the difference between the two pathways. If a DE analysis is performed and only DE genes are included in our model, it will be less likely for us to spot the same structural difference as before because we would lose much group level information by filtering out non-DE genes.

Secondly, not considering the outliers, genes belonging to the JA pathway are mostly concentrated around the origin as well as in quadrant III, meaning that most JA genes are

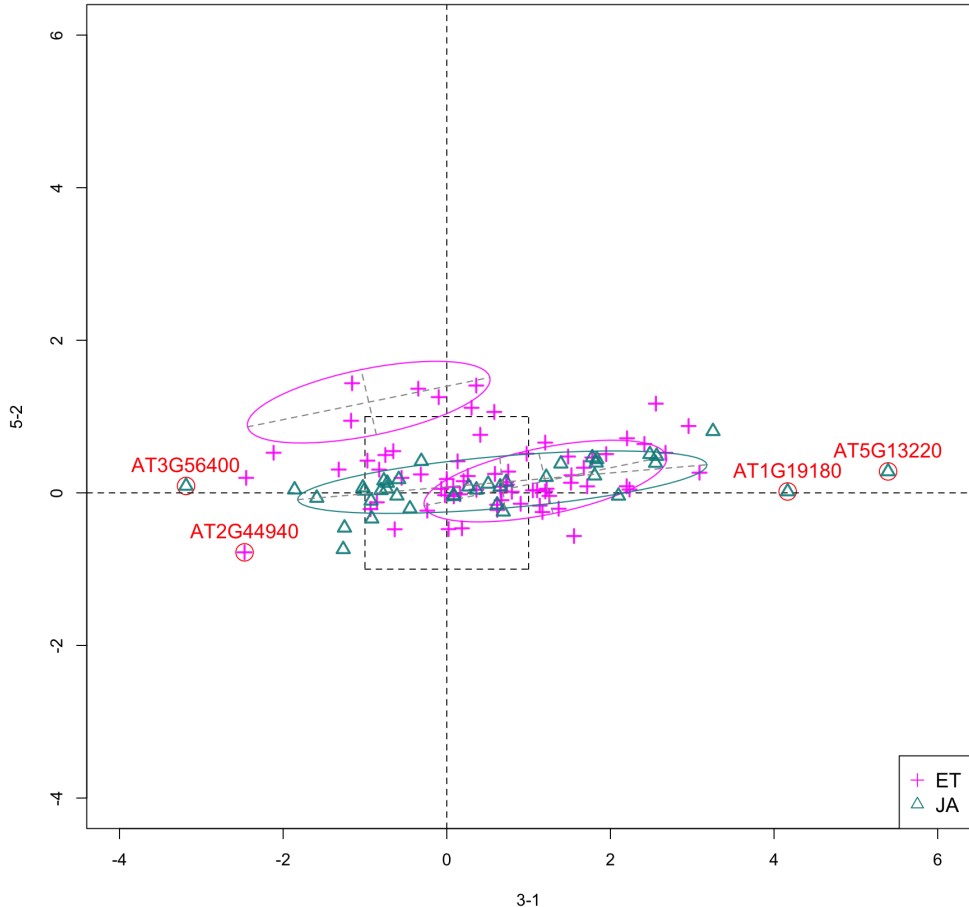

**Figure 5 Scatterplot of data projected on dimensions 3-1 and 5-2.** Pathways are distinguished with color. Ellipses represent estimated covariances fitted by MclustDA. Potential outliers highlighted and labeled with their names. Dashed-line square is ±log(2) range from the origin.

down-regulated under both treatments. The expression pattern of ET pathway genes, however, is more diverse than that of JA genes. These genes populate all four quadrants of the coordinate system, with the highest density in quadrant I followed by quadrant II and III. That is, a majority of ET genes are up-regulated under both treatments, while most of the others are down-regulated under 4–7.

### Top ranked scatterplot: different experiments

In Fig. 5, we show the scatterplot of another top ranked feature combination, [3-1, 5-2], which come from two different experiments.

#### Experiment 3

The focus of this study is the response of *Arabidopsis* to multiple consecutive dehydration stresses (*Ding et al., 2013*). Based on the observation that pre-exposure to abiotic stresses (including dehydration) may alter plants' subsequent responses by improving resistance to future exposures, the researchers hypothesized the existence of "memory

| Table 9 Feature information for experiments 3 and 5. | | |
|---|---|---|
| **Feature ID** | **Sample GEO accession #** | **Description** |
| 3-0 (control) | GSM1173202, GSM1173203 | Watered condition |
| 3-1 | GSM1173204, GSM1173205 | First drought stress |
| 5-0 (control) | GSM1321694, GSM1321704 | Wildtype |
| 5-2 | GSM1321696, GSM1321706 | *atmorc2-1* mutant |

genes": genes that provide altered response to subsequent stresses (*Ding, Fromm & Avramova, 2012*).

A RNA-Seq study is performed to determine the transcriptional responses of *Arabidopsis* plants that have experienced multiple exposures to dehydration stress and compare them with the transcriptional behavior of plants encountering the stress for the first time. The dehydration treatments are applied in the following fashion:

(1) Plants were removed from soil and air-dried for 2 h. Call this exposure Stress 1 (S1).
(2) Plants were then rehydrated for 22 h by being placed in humid chambers with their roots in a few drops of water. Call this step Recovery 1 (R1).
(3) Air-dry R1 plants for 2 h. This is Stress 2 (S2), followed by R2, which is the same as R1.
(4) Air-dry R2 plants for 2 h. This is Stress 3 (S3).

RNA sequencing analyses were then performed on leave tissues from pre-stressed/ watered plants (control), S1 plants and S3 plants. For each treatment group, plants from two independent biological samples were used. In our data, feature 3-1 corresponds to S1, or first drought stress. See Table 9 for a summary.

*Experiment 5*

In this study, the researchers examine the functional relationship between members of the *Arabidopsis* microrchidia (AtMORC) ATPase family (*Moissiard et al., 2014*), which have been shown to be involved in transposon repression and gene silencing. Three of seven MORC homologs were examined: AtMORC1, AtMORC2 and AtMORC6. RNA-Seq experiment using single and double mutants indicates that AtMORC1 and AtMORC2 act redundantly in gene silencing. Wildtype *Arabidopsis* was used as control group. Treatment groups include both single and double mutant lines: *atmorc2-1*, *atmorc2-4*, *atmorc1-2*, *atmorc1-5*, and *atmorc1-2 atmorc2-1*, in which two individual alleles were used for *atmorc1* and *atmorc2*. In our data, feature 5-2 corresponds to the single mutant line *atmorc2-1*. Table 9 includes summary information on this experiment.

*Outliers*

In Fig. 5, we highlighted a few observations considered as outlying, and as before, looked up their information using TAIR. A brief description for each outlier is included in Table 10. Gene **AT3G56400** is again identified as an outlier, mainly because of its highly negative expression level under treatment 3-1, while the near-zero expression level under 5-2 indicates its inactivity under this treatment. Gene **AT5G13220** has the highest

| Table 10 Outlier information for 3-1 and 5-2. | |
| --- | --- |
| Gene name | Description |
| AT3G56400 | WRKY Transcription Factor, Group III; Repressor of JA-regulated genes; Activator of SA-dependent defense genes. |
| AT1G19180 | a.k.a. JAZ1 Nuclear-localized protein involved in JA signaling; JAZ1 transcript levels rise in response to a jasmonate stimulus. |
| AT5G13220 | a.k.a. JAS1, JAZ10 Repressor of JA signaling |
| AT2G44940 | Integrase-type DNA-binding superfamily protein |

expression level under 3-1 among all JA genes, and at the same time not as active under 5-2. This gene is interesting because it functions as a repressor of JA signaling, and its high expression level could be an implication for repression of JA signaling for *Arabidopsis* plants going through first drought stress (3-1).

*Pattern differences*

From the scatterplot, the first thing we can observe is that quite a few genes from both pathways are up- or down-regulated under treatment 3-1, while genes are expressed at an overall low level under 5-2. Nevertheless, a few genes from ET group show overexpression pattern under 5-2. JA pathway genes populate quadrants I, II and III, while ET pathway genes are mainly located in quadrants I, II and IV. Overall, under 5-2, ET genes tends to be more active than JA genes.

## Using Fisher LDA as classifier

With Fisher's LDA as the classifier, we found a distinct assembly of top ranked feature pairs than when MclustDA was used. Table 11 show which 2-subsets produced higher GSI scores than most others.

In Figs. 6 and 7, we show the scatterplot of data projected on to dimensions [3-1, 5-5] and [3-1, 5-9], two of the top ranked feature pairs. Visually, the two pathways groups are not as evidently separated as seen in our previous examples (Figs. 4 and 5).

## GSI for PC transformed data

In Table 12, we report the GSI for PC transformed data, as well as the maximum GSI achieved by subsets of the original data. The proportion of total variation explained is 66.5% for first two PCs, 78.2% for first three, and 85.6% for first four. Through comparison, we observe that using PCs as new features does not necessarily maximize the separation between the distinct groups in the data, therefore confirming our statement in "Dimension reduction via PCA."

## CONCLUSION

In this article, we defined a numeric measure for the separation between different groups of data, and used said measure to perform low dimensional feature subset selection in order to find the most interesting angles to visualize high dimensional data. By applying our method to a multi-experiment RNA-Seq data on *Arabidopsis* leave tissues, we found that the top ranked feature subsets did demonstrate some interesting differences in

**Table 11 Top ranked 2-subsets from multiple runs of 10 × 10 RCV.**

**(a) Run 1**

| Rank | Subset | GSI |
|---|---|---|
| 1 | [3-1, 5-9] | 0.700 |
| 2 | [3-1, 5-5] | 0.688 |
| 3 | [4-4, 4-5] | 0.688 |
| 4 | [2-3, 3-1] | 0.679 |
| 5 | [3-1, 5-3] | 0.675 |

**(b) Run 2**

| Rank | Subset | GSI |
|---|---|---|
| 1 | [3-1, 5-9] | 0.695 |
| 2 | [3-1, 5-5] | 0.687 |
| 3 | [4-4, 4-5] | 0.687 |
| 4 | [3-1, 5-3] | 0.682 |
| 5 | [3-1, 4-5] | 0.678 |

**(c) Run 3**

| Rank | Subset | GSI |
|---|---|---|
| 1 | [3-1, 5-9] | 0.704 |
| 2 | [3-1, 5-5] | 0.693 |
| 3 | [4-4, 4-5] | 0.688 |
| 4 | [3-1, 5-3] | 0.683 |
| 5 | [3-1, 5-1] | 0.675 |

**(d) Run 4**

| Rank | Subset | GSI |
|---|---|---|
| 1 | [3-1, 5-9] | 0.700 |
| 2 | [4-4, 4-5] | 0.689 |
| 3 | [3-1, 4-5] | 0.680 |
| 4 | [3-1, 5-3] | 0.680 |
| 5 | [3-1, 5-5] | 0.680 |

**(e) Run 5**

| Rank | Subset | GSI |
|---|---|---|
| 1 | [3-1, 5-9] | 0.696 |
| 2 | [3-1, 5-5] | 0.693 |
| 3 | [4-4, 4-5] | 0.685 |
| 4 | [2-3, 3-1] | 0.683 |
| 5 | [3-1, 4-5] | 0.679 |

**Note:**
Ties are marked with asterisk (*). Combinations appearing in all five runs are highlighted with distinguishing colors.

expression patterns between two biological pathways, which shows that our method can be a potentially powerful tool in the exploratory analysis of such high dimensional integrated/assembled data from various sources.
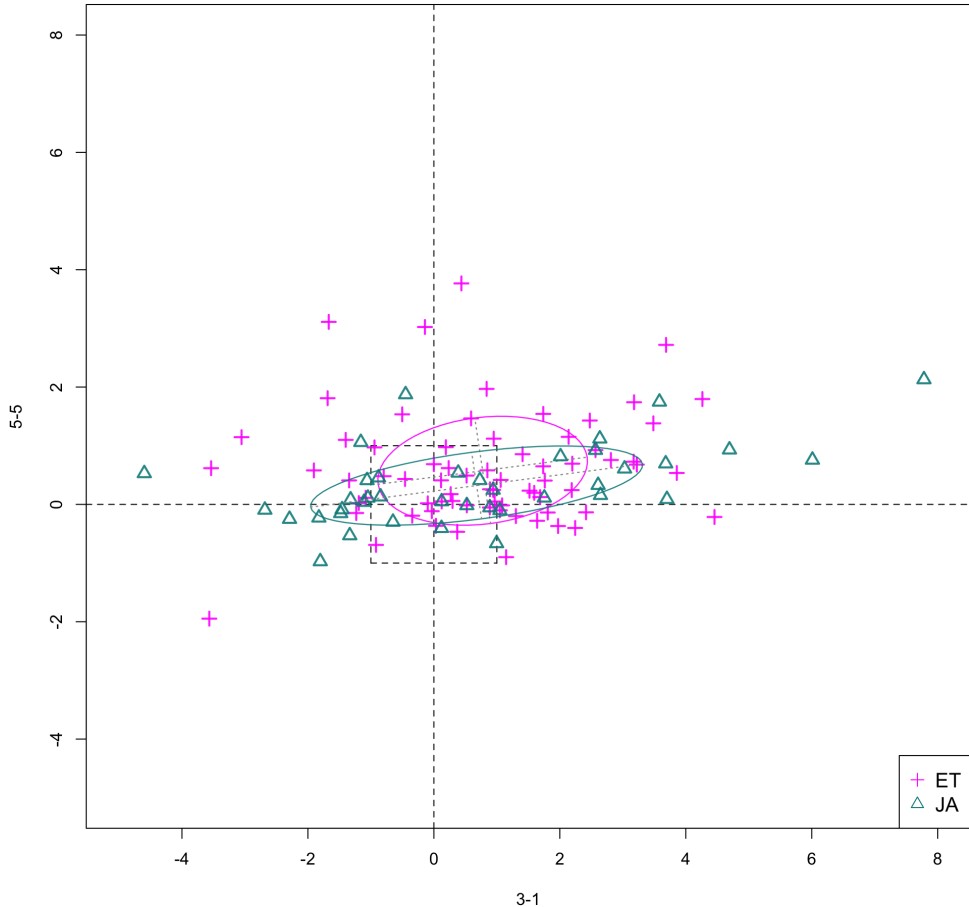

**Figure 6 Scatterplot of data projected on dimensions 3-1 and 5-5.**

## Significance of work

Firstly, our method yields well documented results. We enumerated the GSI for every low dimensional feature subset, and constructed the scatterplots/scatterplot matrices for each case. If scientists know beforehand which features are of interest, they will be able to directly access the corresponding entry in our result. Secondly, through the application of mixture DA, we were able to summarize the expression pattern of groups of genes using a mixture of only a handful of normal components. Furthermore, using the fitted MclustDA ellipses as visual aid, we were able to clearly show the geometric structure of each group and make comparisons. Finally, as seen in Fig. 4, through visualization of the unfiltered data, we are able to identify difference in expression patterns of non-DE genes between two biological pathways.

## Limitations and future work

A limitation of our method is the difficulty of scaling our feature selection method to data of higher dimensions. The first concern is the heavy computational burden required for RCV. In our implementation, although we used parallel computing to speed up

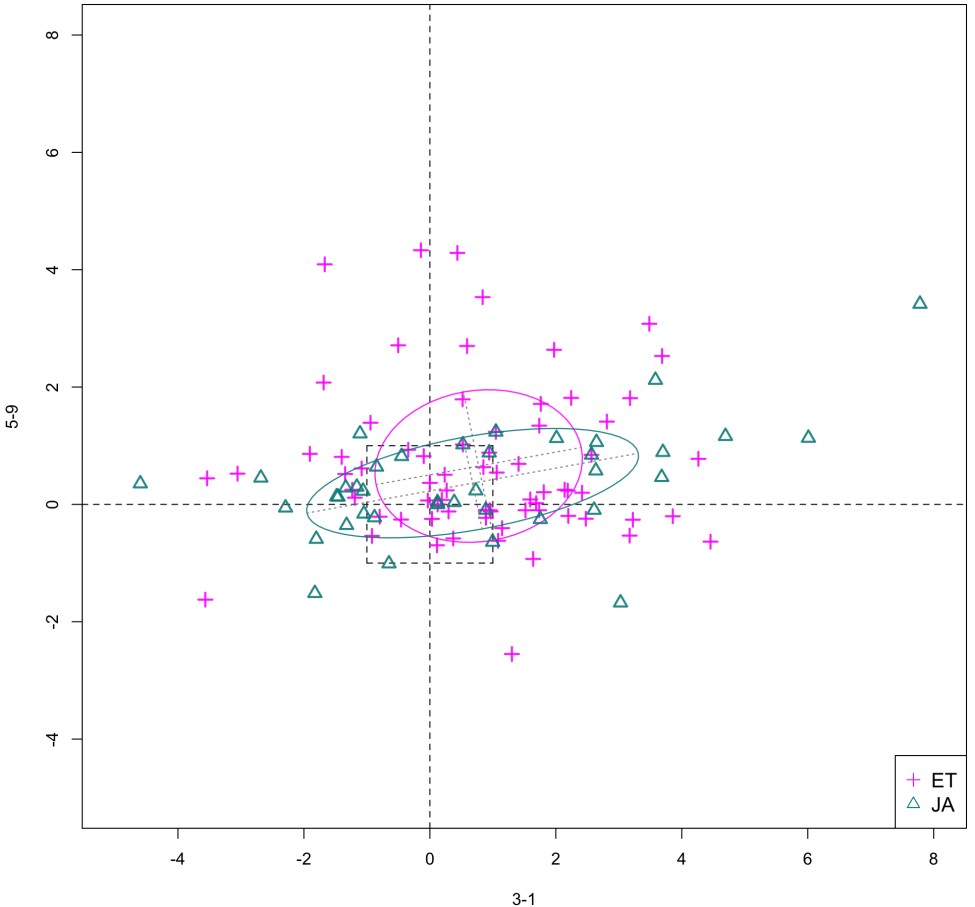

**Figure 7  Scatterplot of data projected on dimensions 3-1 and 5-9.**

**Table 12  Separation index for PC transformed data and maximum GSI for original data.**

| # of PCs | GSI | Features | Max GSI achieved |
| --- | --- | --- | --- |
| 2 | 0.638 | 2 | 0.708 |
| 3 | 0.642 | 3 | 0.736 |
| 4 | 0.639 | 4 | 0.745 |

**Table 13  Average running time for 10-fold cross-validation for all feature subsets, averaged over 50 runs with different random number seeds.**

| Subset dim. | # of subsets | Avg. runtime (s) |
| --- | --- | --- |
| 2 | 253 | 65.04 |
| 3 | 1,771 | 512.61 |
| 4 | 8,855 | 2,241.43 |

computation as much as possible, the actual running times for three- and four-dimensional subset are not quite satisfactory (Table 13), mainly due to the large number of possible subsets. However, in practice, the 2-subset results are usually more interpretable and visually
appealing than its higher dimensional counterparts. Therefore, we recommend doing only two-dimensional feature subset selection for exploratory purposes. In "Repeated cross-validation with MclustDA," we singled out two of the top ranked scatterplots for discussion. Interested readers are directed to the appendix for additional scatterplots and scatterplot matrices for top ranked 3- and 4-subsets (Figs. S1–S8).

Moreover, as pointed out by one of the reviewers, as the sample size increases, one can hope that the bias of the error (hence GSI) estimation will reduce and the ranking of feature subsets from each CV run will become more stable. One can then save computational time by eliminating the repetition, or even the use of CV in the calculation of GSI. In the Appendix, we illustrated how the bias in error estimation will change and how the variability of GSI values from repeated CV will decrease as the sample size increases using simulated data.

Another reason is that the scatterplot matrix becomes less informative when the number of displayed dimensions exceeds four. Even in our study, scatterplot matrices of dimensions 3 and 4 cannot fully reflect geometric properties of the data. For three- and four-dimensional angles, the scatterplot matrix only shows projections to all axial dimensions, which doesn't precisely convey the amount of separation between two classes, computed using all three or four dimensions. It is difficult to visualize the geometric and topological differences by only looking at individual panels of scatterplots. To more effectively visualize higher dimensional feature subsets, we can consider using interactive visualization tools, such as GGobi (*Swayne et al., 2003*) and R Shiny (*Chang et al., 2017*). Both tools allow users to identify the same point in all panels of a scatterplot matrix, significantly increasing its visual expressiveness.

### Error rate definition

In our definition of TPM in (1), we made the assumption that the cost of misclassifying an object from either class is the same, i.e., $c(1|2) = c(2|1)$. We can adjust the cost values if we are more concerned about correctly classifying a certain class of observations.

### Evaluating reproducibility of experiments

Currently, a typical DE analysis is conducted in a gene-wise manner, that is, genes are treated as observations and the treatment conditions as features. In our study, we took the same approach because our goal was to differentiate expression pattern between two groups of genes. However, with the increase in the availability of RNA-Seq data thanks to advances in information technology, we can also study the comparability and reproducibility of RNA-Seq experiments. In this sense, we will be exploring the relationship between treatment conditions or experiments, with genes acting as features/variables. Evaluation of experiment reproducibility is usually accomplished by performing the same experiment using the same setting, which is, unfortunately, not a common practice in RNA-Seq studies. In light of this, one of our long-term goal is the quantification of similarity between RNA-Seq experiments, which not only accounts for differences in experimental designs and parameter settings, but also utilize the information hidden in the expression of genes.

## Significance evaluation

We were advised on whether we can assign a measure of significance for the top ranked feature set found by our method. Our interpretation of "significance" in this context is as follows: Under the assumption that the feature pair contain no group information on the genes, what is the probability of obtaining a GSI value as high as or higher than the value computed by our method (i.e., a $p$-value)? We can use a permutation test for this purpose, by following these steps: (1) For a particular feature pair, randomly reassign group labels to genes, (2) Calculate GSI value using the permuted data, (3) Repeat one and two for $M$ (usually large) times, and establish the distribution of GSI value, and (4) Calculate empirical $p$-value based on this distribution. Due to time constraint, we found such a $p$-value for only one top ranked feature pair, [4-7, 4-8], by calculating $M = 2,000$ GSI values from permuted data, and obtained the empirical $p$-value of 0.0055, indicating the significance of this pair of features.

## Handling effect and CV

As shown in "Quantification of group separation," CV was used for the calculation of MER. A recent study (*Qin, Huang & Begg, 2016*) points out that in the existence of handling/batch effect of objects to be classified, CV will often underestimate the error rate, mainly due to the complete or partial confounding between batches and subject classes. In our study, the biological samples were indeed handled separately under different circumstances, which means the treatment-control contrasts derived using these samples are also subject to an inherent group structure. However, rather than classifying the biological samples or contrasts, this study focuses on classifying genes to one of two pre-specified gene pathway groups. Since a common assembly pipeline was used to process the data, the list of genes should be considered as uniformly handled, and therefore free from the handling/batch effect discussed in the aforementioned article. The batch effect will, on the other hand, affect the covariance structure of the space of log fold changes that we explore, which will be reflected when Mclust models are fitted to the lower dimensional subspaces.

# ACKNOWLEDGEMENTS

The authors gratefully acknowledge Jeff Chang, Sarah Emerson and Duo Jiang for their valuable insight and comments, and Bin Zhuo for the collection of data and the discovery of a set of stably expressed reference genes.

## Funding

The authors received no funding for this work.

## Competing Interests

The authors declare that they have no competing interests.

## Author Contributions

- Wanli Zhang conceived and designed the experiments, performed the experiments, analyzed the data, contributed reagents/materials/analysis tools, prepared figures and/or tables, authored or reviewed drafts of the paper, approved the final draft.
- Yanming Di conceived and designed the experiments, performed the experiments, analyzed the data, contributed reagents/materials/analysis tools, authored or reviewed drafts of the paper, approved the final draft.

## Data Availability

   GitHub: https://github.com/wzhang43/High-Dimensional-Data-Visualization.

## Supplemental Information

Supplemental information for this article can be found online at http://dx.doi.org/10.7717/peerj.5199#supplemental-information.

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
