# Peer review of "Searching for best lower dimensional visualization angles for high dimensional RNA-Seq data"

_PeerJ, doi:10.7717/peerj.5199_

## Round 0.1 · original submission · Major Revisions

The reviewers made some good suggestions to strengthen the paper. Hope to see the revised version soon.

Reviewer 1 ·

Basic reporting

Table 1: Add additional columns to provide a clear description of the study designs. For example, profiling platform, sample size, groups under comparison, etc.

Paragraph below Table 1: Please state the method for normalization and the method for differential expression, so that the readers don't have to refer to the Zhuo et al 2016 paper.

Experimental design

When fitting MclustDA model, how is the number of clusters chosen?

It has been recently reported in the literature (PMID: 27601553) that, when molecular data possess confounding handling effects and normalization is used to the data, cross-validation tends to under-estimate the error rate. As a result, the repeated CV method (that the authors use for estimating error rates to rank gene combinations) may be prone to this issue as well. Would there be a way to check its performance, or to use an alternative method that is more robust to handling effects or data normalization?

Validity of the findings

A method called 'top scoring pairs' (R package tspair) also looks at the problem of ranking gene pairs with regarding to the classification power. How would the proposed method compare to this method, when the goal is to choose top gene pairs?

The proposed method is rather computationally involved and intensive. Is there an R package available for others to use the proposed method to reproduce the results and to apply it to other datasets?

Reviewer 2 ·

Basic reporting

no comment

Experimental design

no comment

Validity of the findings

no comment

Additional comments

Better visualization is important for biologists to interpret RNA-seq data. In this paper, the author proposes a method for selection sets of two, three, or four genes, on which samples can be best separated into pre-specified groups. The problem focused on is biologically meaningful, and I am expected a good method, like what the author has proposed, will be very helpful and widely adopted by biologists. The proposed method is intuitive, easy to understand, and statistically solid. The paper is well organized and clearly written. I have one major comment that the author needs to address, and two suggestions, which I think would further strengthen the paper, that the author may choose to adopt.

Comment:

1. The major method the author compared their method with is PCA (in sections 3.4 and 4.2), and indeed PCA does not use group labels and thus one would not expect it to work well. However, the author should also compare with Fisher’s LDA, which is based on a much simpler model but also computationally much quicker, and see how much benefit of performance (in the sense of finding better gene sets) the proposed method has brought.

Suggestions:

1. If the author can make the proposed method computationally faster, the biologists, who are likely to be the main users of this method, will favor the method more. The current method requires pre-selection of a subset of genes that are likely relevant based on known pathway information. Particularly, they choose no more than 100 genes for their real data examples. If this pre-selection step can be eliminated or loosened, the method will likely be more widely used. One possible way of accelerating the algorithm is to simplify the cross-validation step. The author has been cautiously enough to do K-fold CV with C repetitions, which is good in the sense of making the results stable. However, I guess the repetitions of CV, or even the whole CV, can be eliminated, at least in two cases: (1) when the number of samples (RNA-seq experiments) are not too small; (2) for sets of (two, three, or four) genes when a non-significance is clear. For the first case, the author may construct computational study and see what number of samples is not “too small”. For the second case, sets of genes that are obviously non-significant (by a single-round calculation without CV or by other intuitive methods) should be eliminated. Other ways to accelerate the algorithm is also possible. Further, I would suggest the author focus one finding sets of two genes, as multiple 3-D or 4-D scatterplots are rarely used anyway.

2. Currently, the method gives a GSI value for each identified set of genes. It will be helpful to also give some sort of estimated p-values or FDRs for each identified set. This estimation does not need to be accurate; even a rough estimate will be helpful to the users.

---

## Round 0.2 · Minor Revisions

Please address the remaining comment from Reviewer 1.

Thanks,

Xiangqin

Reviewer 1 ·

Basic reporting

None

Experimental design

The authors response to the second comment under 'Experimental design' is not satisfactory. They reasoned that they 'classify genes instead of biological samples'. However, the purpose of their gene-selection method is to classify samples, and uses CV for the optimization of the selection. The authors should, at the very least, discuss the potential caveat of using CV in molecular data that may possess confounding handling effects.

Validity of the findings

None

Additional comments

None

Reviewer 2 ·

Basic reporting

no comment

Experimental design

no comment

Validity of the findings

no comment

Additional comments

The author has addressed all my concerns in full.

---

## Round 0.3 · accepted · Accept

Thanks for the revision. Your paper is now accepted.

#